# Use Cases Requiring Privacy-Preserving Record Linkage in Paediatric Oncology

**DOI:** 10.3390/cancers16152696

**Published:** 2024-07-29

**Authors:** Dieter Hayn, Karl Kreiner, Emanuel Sandner, Martin Baumgartner, Bernhard Jammerbund, Markus Falgenhauer, Vanessa Düster, Priyanka Devi-Marulkar, Gudrun Schleiermacher, Ruth Ladenstein, Guenter Schreier

**Affiliations:** 1Center for Health and Bioresources, AIT Austrian Institute of Technology, 8020 Graz, Austriamartin.baumgartner@ait.ac.at (M.B.);; 2Institute of Neural Engineering, Graz University of Technology, 8010 Graz, Austria; 3St. Anna Kinderkrebsforschungs GmbH, 1090 Wien, Austriaruth.ladenstein@ccri.at (R.L.); 4Institut Curie, 75005 Paris, Francegudrun.schleiermacher@curie.fr (G.S.)

**Keywords:** privacy, record linkage, paediatric oncology, data management, secondary use, European Health Data Space (EHDS), General Data Protection Regulation (GDPR)

## Abstract

**Simple Summary:**

Large datasets concerning childhood cancers are rare. Therefore, it is important to fully exploit all available data, which are distributed over several resources, including biomaterials, images, clinical trials, and registries. With privacy-preserving record linkage (PPRL), datasets can be merged, without disclosing the patients’ identities. Although PPRL is already implemented or described in various settings, use case descriptions are fragmented and incomplete. The present paper gives an overview of current and future use cases of PPRL in childhood cancer. We screened the literature, projects, and trial protocols, analysed a hypothetical patient journey, and discussed use cases with experts. All the identified use cases were structured along six key dimensions. We conclude that PPRL is a key concept in childhood cancer. Therefore, PPRL strategies should already be considered when starting research projects, to avoid distributed data silos, to maximise the knowledge derived from collected data, and, ultimately, to improve outcomes for children with cancer.

**Abstract:**

Large datasets in paediatric oncology are inherently rare. Therefore, it is paramount to fully exploit all available data, which are distributed over several resources, including biomaterials, images, clinical trials, and registries. With privacy-preserving record linkage (PPRL), personalised or pseudonymised datasets can be merged, without disclosing the patients’ identities. Although PPRL is implemented in various settings, use case descriptions are currently fragmented and incomplete. The present paper provides a comprehensive overview of current and future use cases for PPRL in paediatric oncology. We analysed the literature, projects, and trial protocols, identified use cases along a hypothetical patient journey, and discussed use cases with paediatric oncology experts. To structure PPRL use cases, we defined six key dimensions: distributed personalised records, pseudonymisation, distributed pseudonymised records, record linkage, linked data, and data analysis. Selected use cases were described (a) per dimension and (b) on a multi-dimensional level. While focusing on paediatric oncology, most aspects are also applicable to other (particularly rare) diseases. We conclude that PPRL is a key concept in paediatric oncology. Therefore, PPRL strategies should already be considered when starting research projects, to avoid distributed data silos, to maximise the knowledge derived from collected data, and, ultimately, to improve outcomes for children with cancer.

## 1. Introduction

Over half a century, consecutive clinical trials have improved survival rates across the spectrum of paediatric oncology (PO). Nevertheless, more than 6000 children die from cancer in Europe each year [1]. Since large datasets in PO are inherently rare, it is paramount to fully exploit all available data. Current PO research data are distributed over several resources holding biomaterials and their respective analysis, various image modalities, and clinical trial and registry data. In addition to clearly defined research goals in prospective settings, secondary use of data opens up important research and knowledge generation opportunities. However, secondary use is associated with various hurdles, including technical issues, legal barriers due to increasingly complex regulations, and ethical discussions about explicit informed consent for each analysis as compared to broad or even opt-out consent. Due to these and other reasons, most data are currently locked up in data silos.

The following levels of privacy can be distinguished: Personalised data, containing personal unique identifiers (IDs) or quasi-identifiers (QIDs), such as name, data of birth, and others, as defined by [2].Pseudonymised data, containing patient codes (“pseudonyms”) that can be associated with related QIDs, if necessary.Anonymised data, which cannot be associated with related QIDs anymore.Aggregated data, e.g., statistical results as published in journals.

According to the General Data Protection Regulation (GDPR) [3], whenever possible, research data need to be collected in a pseudonymised way. As such, in general, different resources must generate different pseudonyms for one and the same patient. To overcome barriers due to distributed (pseudonymised) data, the European Joint Programme on Rare Diseases (EJP RD) is currently creating an effective rare diseases research ecosystem (https://www.ejprarediseases.org/ (accessed on 23 July 2024)). The EJP RD is based on the FAIR (Findable, Accessible, Interoperable, and Re-usable) principles [4], which are key to exploit existing data and support big data and artificial intelligence applications, as recently demanded by the European Society for Paediatric Oncology (SIOPE) [5,6].

Privacy-preserving record linkage (PPRL) concerns the linking of personalised or pseudonymised datasets, without disclosing the patients’ identities. PPRL is a key concept for PO research [7]. During pseudonymisation, PPRL can avoid unintended double registration, which would lead to empty or fragmented pseudonymised records and—if followed by double randomisation—unbalanced cohorts.

PPRL is required for (a) merging patients from contexts with identical variables and/or (b) merging different data types from multiple sources holding the same patients (Figure 1).

Most PPRL publications focus on the technical implementation of PPRL algorithms. In general, PPRL algorithms can follow phonetic coding-based (e.g., [8,9]), hashing-based (e.g., [10,11]), reference-value-based (e.g., [12,13]), embedding-based (e.g., [14,15,16]), differential privacy-based (e.g., [17]), or secure multiparty computation-based techniques, which can either be applied on QIDs or on the clinical data themselves (e.g., [18,19,20,21,22]). Some solutions only support linkage in case of perfectly matching records. More comprehensive solutions can also link slightly differing records, e.g., in case of typing errors or missing data. Since both false positive as well as false negative linkage may have severe consequences, the optimal threshold of accordance must be chosen depending on the use case.

To compare PPRL solutions, Vatsalan et al. [23] and Gkoulalas-Divanis et al. [24] published PPRL overview papers, each with one out of many dimensions relating to application areas. A comprehensive overview of various aspects of PPRL was published by Christen et al. [25]. Their introduction summarises several aspects of why data should be linked at all and which sources of data might be linked, and they describe selected use cases. In particular, use cases focusing on health service research, national health insurance data in Germany, official statistics, and longitudinal studies are considered herein. In the conclusions of three recently published whitepapers [26,27,28], the European Union Agency for Cybersecurity (ENISA) highlights the importance of pseudonymisation and PPRL in distributed research environments as well as the need for cooperation in terms of implementation, development of application scenarios, regulatory support, and dissemination of best practice models. The use cases described especially in [26] represent a valuable resource for the present paper. PPRL is also expected to play a major role in the upcoming European Health Data Space (EHDS), which is supposed to provide a framework for structured and systematic secondary use of health data in the EU on a large scale [29].

For PO in Europe, there are mainly two PPRL services currently available: (1) SPIDER (https://eu-rd-platform.jrc.ec.europa.eu/spider/ (accessed on 23 July 2024)) focusses on PPRL between primary sources holding QIDs with perfect matches only (no typing errors, etc.). (2) The European Patient Identity (EUPID) Services [30] were designed to also link pseudonymised resources. EUPID further supports PPRL in case of typing errors based on phonetic hashing algorithms. The EJP RD virtual platform currently supports EUPID.

PPRL has been implemented in various PO settings. However, the related use cases have been described implicitly, based on the requirements of the respective projects. Use cases specified in the literature are fragmentary and spread over various publications. Therefore, the present paper provides a comprehensive overview of current and future use cases for PPRL in PO.

## 2. Materials and Methods

We followed three strategies to compile a comprehensive overview of PPRL use cases in PO.

First, we analysed the literature, existing PO projects, and trial protocols concerning how and why they implemented PPRL. For the literature search, we applied the following search terms in PubMed:


*((((“record” OR “dataset” OR “registr*”) AND (“link*” OR “merg*” OR “combin*”)) AND (“privacy” OR “GDPR” OR “data protection” OR “pseudonym*” OR “anonym*” OR “de-identif*” OR “deidentif*” OR “leak*”)) OR “PPRL”) AND ((“paediatr*” OR “pediatr*” OR “child*” OR “infant”) AND (“cancer” OR “oncolog*” OR “tumor” OR “tumour”)).*


Titles and abstracts and, in a second step, full-text publications were manually filtered to include only papers containing information concerning use cases of PPRL in PO, and the relevant use cases were extracted. The search terms revealed 85 articles, of which 23 articles containing use cases of PPRL in PO were considered.

Secondly, inspired by a recent paper by Ly et al. [31], we identified use cases along a hypothetical patient journey, as shown in Figure 2.

Finally, we discussed the identified use cases with stakeholders from (a) SIOPE and (b) the EJP RD to complement the list and to firm up the single use cases.

The identified use cases were then analysed and categorised along key dimensions.

## 3. Results

### 3.1. Overview of Dimensions in Privacy-Preserving Record Linkage

During our research, we found that a one-dimensional list of use cases is not feasible due to the variety of different PPRL applications in PO. Therefore, we categorised use cases along key dimensions, as illustrated in Figure 3.

Section 3.2 describes single dimensions of Figure 3 (“horizontal” slices), i.e., distributed personalised/pseudonymised data sources, record linkage/linked data, and data analysis, including PO examples.

Section 3.3 summarises multi-dimensional use cases (“vertical” paths through Figure 3), based on the EJP RD use case specification format: “As a <stakeholder>, I would like to <research focus> so that I can <objective> and visualise the result as a <output format>” (Terms between “<” and “>” were replaced by use-case-specific contents).

### 3.2. Single-Dimensional Use Cases

#### 3.2.1. Distributed Personalised/Pseudonymised Data Sources

This chapter summarises the rationale for why PO data are distributed not only in routine care but also in research-related data environments.

##### Multiple Hospitals

Patients might be treated in more than one hospital over time, leading to distributed personalised records, thus increasing the risk of double registration.

PO example—Specialised and local hospitals: Patients with complex or rare diagnoses move from local to specialised hospitals. Vice versa, less complex cases are transferred from specialised to local hospitals.

PO example—Second opinion: Due to the severity of paediatric cancer, parents often seek a second opinion from another hospital.

##### Different Types of Data

Pseudonymised research data of different types are often distributed over different sources.

PO example—Data sources in neuroblastoma research: Neuroblastoma trial data are collected in several electronic data capture systems. Additionally, there are national and international registries (e.g., the French National Registry of Childhood Cancers [29]). Bio-sample data are stored in biobanks, some of which can be queried via BBMRI-ERIC [32]. Sequencing results can be stored in the RD-Connect Genome-Phenome Analysis Platform (GPAP) [33]. DICOM images are collected on a pseudonymised image management server.

##### Suspected Diagnosis

Eligibility criteria for clinical trials often require a confirmed diagnosis, although confirming a suspected diagnosis may already provide valuable research data (e.g., tumour samples in a biobank).

PO example—SIOPEN BIOPORTAL: SIOPE Neuroblastoma (SIOPEN) is currently establishing a “SIOPEN BIOPORTAL” (NCT05192980) (https://clinicaltrials.gov/study/NCT05192980 (accessed on 23 July 2024)), which supports registration and data collection based on suspected diagnoses with a specific BIOPORTAL pseudonym. This pseudonym can be used to send biological samples to reference laboratories. After confirmation of the diagnosis, the patient is registered to the corresponding trial (e.g., low/high risk) and a second trial-specific pseudonym is generated.

##### Multiple Subsequent Trials

Throughout their lifetime, patients can be included in one or more trials.

PO example—Poor response: If a high-risk neuroblastoma patient enrolled in an upfront treatment trial such as HR NBL2 (NCT04221035) (https://clinicaltrials.gov/study/NCT04221035 (accessed on 23 July 2024)) did not respond to the induction chemotherapy, the patient was transferred to an even more intensified treatment scheme, e.g., VERITAS (NCT03165292) (https://clinicaltrials.gov/study/NCT03165292 (accessed on 23 July 2024)).

PO example—Relapse: In case of a relapse, patients previously treated in a primary tumour study were frequently enrolled to relapse studies, such as BEACON (NCT02308527) (https://clinicaltrials.gov/study/NCT02308527 (accessed on 23 July 2024)). Subsequently, patients could further be enrolled to national or international precision medicine programs or phase 1 or 2 Innovative Therapies for Children with Cancer (ITCC) trials like ESMART (NCT02813135) (https://clinicaltrials.gov/study/NCT02813135 (accessed on 23 July 2024)).

##### Multiple Concurrent Trials

There are trials that allow for or even foresee the participation in other trials at the same time.

PO example—OMS trial: The Opsoclonus Myoclonus Syndrome (OMS) trial (https://clinicaltrials.gov/study/NCT01868269 (accessed on 23 July 2024)) collects data concerning OMS, a rare syndrome in children that appears frequently in patients with neuroblastoma. Neuroblastoma patients may additionally be treated according to neuroblastoma trial protocols, where oncological data are collected. Therefore, the OMS infrastructure links OMS-specific trial data with oncological data from different neuroblastoma trials [34].

##### Transition from a Trial to Long-Term Follow-Up

Long-term follow-up data are essential but often not collected in trial databases or only for short periods.

PO example—Long-term follow-up: Since PO patients are frequently treated in more than one trial, long term follow-up data should be preserved for all trials, even for decades, without re-entering data in each context. The SIOPEN BIOPORTAL envisages reporting the long-term follow-up data within a neuroblastoma registry, which includes core, patient, tumour, and epidemiological data, with the aim to operate for longer periods.

##### Transition from Routine Care to Research (and Back)

For any trial patient, specific data are collected in routine care and others in research infrastructures, requiring transition from one to the other.

PO example—Continuity of care: With improving treatment options, continuity of care for PO survivors beyond cancer treatment is becoming increasingly important. SIOPE is working on guidelines dealing with the transition from paediatric patients to adolescent or adult survivors [35].

PO example—Late effects: The majority of the nearly 500,000 PO survivors in Europe experience late effects, with increasing incidence over the lifetime and impacts on their daily life [36]. Identification of late effects, even decades after the treatment, and development of long-term follow-up guidelines require transitions from research to routine care and back.

PO example—End-of-life treatment: Henson et al. used linked data to analyse prescriptions dispended in the community during end-of-life care of cancer patients [37].

#### 3.2.2. Record Linkage/Linked Data

##### Temporarily Linked Data

Based on related pseudonyms of distributed contexts, data are temporarily linked in a central service, specific calculations are applied to the linked data, and the result is returned to the user.

PO example—Counting shared patients: Via the EJP RD virtual platform, users will be able to count patients that are registered in two different contexts, such as the HR-NBL1 trial and the International Neuroblastoma Risk Group (INRG) databases.

##### Linked Data Stores and Registries

Linked data can be stored in specific data stores, either to support subsequent tasks or to serve as a long-term data source (e.g., a linked registry). Therefore, regular updates must be supported, including the addition of new data, completion of previously missing data, or correction of data.

PO example—INRG database: Based on agreements between SIOPEN and other research organisations (e.g., US Children’s Oncology Group, COG), core PO trial data are submitted to the INRG-DB [38] after publication. The HR-NBL1 trial [39] was amended several times. Over the years, different randomisations were amended, and the corresponding results were successively published. Some patients were randomised more than once (e.g., pre-induction and pre-maintenance). Data of such patients were sent to the INRG-DB once per randomisation, and PPRL ensured that such patients were not registered twice.

PO example—The international diffuse intrinsic pontine glioma registry [40] provides an infrastructure for acquisition of biological specimens, imaging, and correlative clinical and genomics data to facilitate basic and translational research studies in diffuse intrinsic pontine glioma from 55 collaborating institutions in the United States, Canada, Australia, and New Zealand, including clinical, demographic, radiologic, and pathologic data.

#### 3.2.3. Data Analysis

##### Counts, Statistical Measures, and Statistical Plots

Counting linked data without counting any patient twice and/or including filters (e.g., age < 3 years) requires PPRL. Statistical analyses can be applied on linked data, providing specific measures (e.g., mean value) or statistical plots (e.g., Kaplan–Meier curve).

PO example—Long-term toxicity: Calculate the rate of toxicity in patients who have been treated with a specific drug (e.g., cisplatin) and had a specific symptom thereafter (e.g., ototoxicity) in late-effect research scenarios. Or evaluate the effect of cancer treatment on paternity through use of assisted reproduction technology [41].

PO example—Large-scale statistics: Various studies applied PPRL to perform large-scale statistical analyses on cancer data, e.g., to analyse long-term survival after paediatric cancer [42] or after cancer in general (including PO) [43], to apply benchmarking on paediatric cancer survival [44], to develop a national resource of patient-level genomics laboratory records including PO patients [45], or to accelerate research on pontine glioma [40]. Walker et al. applied PPRL to analyse the prevalence of PO and other conditions increasing the risk of severe COVID-19 disease [46]. Finally, PPRL was applied to determine the incidence of melanoma [47] and of childhood cancer in general [48].

PO example—Effect of specific parameters on PO incidence: Analyse the effect of specific parameters from one dataset on the PO incidence as documented in another source (see, e.g., [49,50,51,52,53,54]).

PO example—Healthcare management: Use linked datasets for cost effectiveness analyses [55], data quality assessment [43], benchmarking [56], or capacity building [57].

##### Tables with Subsets of Linked Data

Subsets of linked data are provided to the user.

PO example—List of follow-ups of a patient: Provision of a table listing all follow-ups of a specific patient, documented in different data sources, including date of follow-up, originating data source, and the documented status of the patient.

##### Single Case Visualisation

Distributed data of a single patient are comprehensively visualised, either in a personalised (healthcare) or pseudonymised (research) context.

PO example—Tumour boards: Tumour boards enhance cancer treatment strategies. In the realm of PO, these boards are frequently conducted in a virtual setting, facilitating the collaboration of external experts with local physicians (see, e.g., [58]). Consequently, it is advisable to share only pseudonymised data that have emerged from both routine care (e.g., images) and research (e.g., randomisation arms).

##### Patient Apps and Digital Companions

Patient apps and digital companions represent a powerful way to provide outcomes of PPRL not only to researchers but also to patients.

PO example—Survivorship passport: The survivorship passport links core treatment data with follow-up recommendations based on guidelines [59]. It will closely be linked to electronic medical records and to national cancer and survivorship registries, supporting regular updates and generation of care plans. The implementation of the survivorship passport is currently being piloted in six European countries [60].

##### Artificial Intelligence (AI)

To tap the full potential of AI, PPRL of distributed data sources is essential.

PO example—Image-defined risk factors: Image-defined risk factors (tumour size, anatomical regions, etc.) are key for risk-adjusted treatment of cancer. However, raw images either remain in the local centre’s Picture Archiving and Communication System (PACS) or they are separately stored on specific image management servers. In the PRIMAGE project [61], DICOM-formatted raw image data from a PO image management server were integrated with clinical trial data and stored on the PRIMAGE platform. AI was applied to predict outcomes from the images.

### 3.3. Multi-Dimensional Use Cases

Table 1 presents a selection of identified high-priority use cases following specific “vertical” paths along the dimensions illustrated in Figure 2, represented in the format developed in the EJP RD project.

## 4. Discussion

PO stands to benefit significantly from PPRL due to the scarcity and distribution of data, multimodal treatments, and complex patient journeys. This paper provides a comprehensive overview of potential use cases, emphasising that the list is not exhaustive and new dimensions may emerge in the future. While focusing on PO, most aspects discussed are also applicable to other conditions, rare diseases in particular.

Most identified use cases concern the analysis of clinical outcomes (long-term toxicities, event-free survival, overall survival) and the effect of certain parameters on the incidence/prevalence of PO. However, we identified a wide range of use cases, which require PPRL in PO care and/or PO research.

Many identified use cases concerned neuroblastoma research, where a European IT infrastructure supporting record linkage on clinical trial data, biological data from different samples, and imaging data has already been introduced in the FP5 project SIOPEN-R-NET. During the FP7 project ENCCA (2011–2015), the EUPID Services have been developed, which provide a more privacy-preserving approach for linking records. Since then, PPRL services have continuously supported collaborative European neuroblastoma research by providing large, linked datasets, which contribute to significantly improved survival rates as achieved in the past decade.

PPRL is a complex task, which is associated with various risks in terms of usability, costs, privacy preservation, potential misuse, the risk of breaking confidentiality, the risk of mismatched identities, etc. Different technical approaches for realising PPRL have different advantages and disadvantages in all these dimensions. Special care should be taken if PPRL is applied based on QIDs that can change over time (e.g., names in case of marriages, gender, etc.), especially if PPRL is to be supported in long-term settings, such as, e.g., the survivorship passport [59]. The present paper, however, concentrates on use cases, omitting technical details, privacy and security aspects, and consent considerations.

The use cases in this paper were identified in a three-step approach (screening of the literature, projects, and trials, hypothetical patient journey, and discussion with experts). On the one hand, the literature search was conducted by one researcher only. Therefore, use cases might have been overlooked that would have been identified if screening had been performed by more researchers. However, due to the multistep approach, such use cases were most likely identified in another step. On the other hand, even articles that did not cover PPRL and PO were considered, if use cases that are also applicable for PPRL in PO were described.

Selecting the right PPRL solution before starting data collection is crucial, especially for pseudonymised record linkage. The lack of standards for and interoperability between PPRL services is acknowledged, and efforts within EJP RD are underway to address this. References to implementation projects are provided for some use cases, with varying stages of development. The evolution from project design to pseudonymisation and linked data analyses is noted. As applications progress, PPRL is recognised as a key concept of a broader privacy-preserving research landscape. Integration with other approaches like privacy-preserving artificial intelligence and federated/distributed learning is anticipated. For PO in Europe, currently, mainly SPIDER and the EUPID Services are implemented, which should be considered when setting up new projects. Choosing the appropriate privacy-preserving strategy will be a key consideration in future research across diverse fields, not limited to PO.

## 5. Conclusions

Research in PO has been at the forefront of PPRL implementations, making significant strides in realising tangible outcomes from linked records. As PPRL gains prominence across various application domains, both existing and new use cases are anticipated to evolve. The long-term nature of underlying diseases underlines the importance of adopting a robust PPRL strategy early on. By doing so, the benefits of linked data can be harnessed sooner, ensuring that datasets collected without a PPRL concept do not miss out on future applications and fall short of maximising patient benefits.

## Figures and Tables

**Figure 1 cancers-16-02696-f001:**
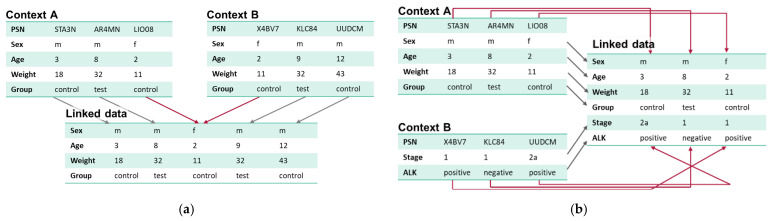
(**a**) Merging patients from contexts with identical variables mainly focuses on the identification of duplicate patient records in different contexts. (**b**) Merging different data types from multiple sources holding the same patients focuses on the identification of related patient records in different contexts.

**Figure 2 cancers-16-02696-f002:**
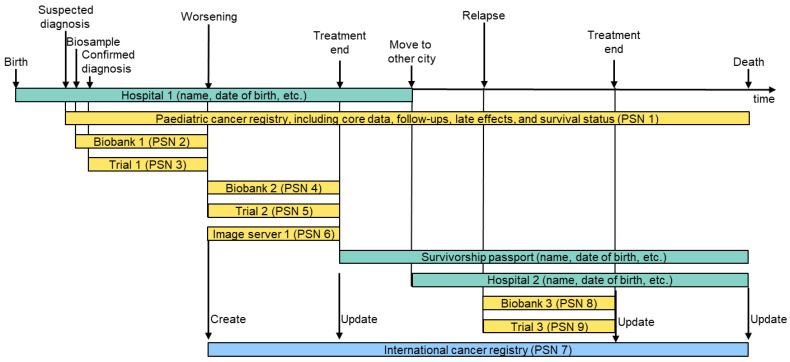
Hypothetical patient journey and related data sources. In this example, a patient is diagnosed with a primary tumour that worsens during treatment in trial 1. Treatment in a second trial 2 is successful. The cancer survivor suffers from a relapse a certain time after healing, which can again be treated successfully. In this patient journey, we end up with nine different pseudonyms (PSNs) for one patient. Green: personalised data; yellow: pseudonymised data; blue: linked data.

**Figure 3 cancers-16-02696-f003:**
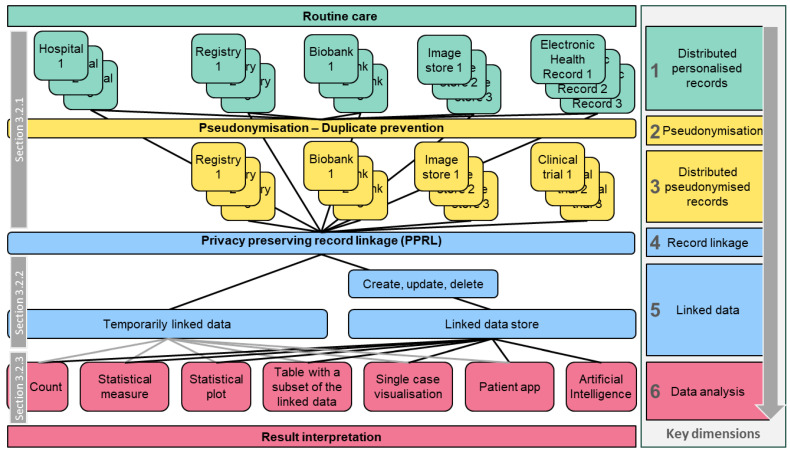
Overview of data residing in personalised routine care and different pseudonymised contexts, illustrating how data can be linked and how the linked data can be analysed to obtain new knowledge. On the right-hand side, the relation to the respective six key dimensions of record linkage is shown. Grey boxes reference chapters describing single-dimensional use cases.

**Table 1 cancers-16-02696-t001:** Subset of possible use cases per stakeholder requiring privacy-preserving record linkage identified as potentially useful in paediatric oncology (PO). The use cases are displayed in the EJP RD use a case specification format, i.e., “As a <stakeholder>, I would like to <research focus> so that I can <objective> and visualise the result as a <output format>”.

Stakeholder “As a...”	Research Focus “...I Would Like to...”	Objective “...so That I Can...”	Output Format “...and Visualise the Results as...”
clinician (corresponds to PO patient OR parent of a PO patient)	know if there have been patients in the past in any source who were similar to my patient, and their outcome	choose the optimal therapy for my patient	a table with key patient characteristics, data sources, and outcomes
	discuss my patient’s case in a tumour board based on all the distributed data of my patient		comprehensive overview of my patient’s case including tables, images, etc.
	compare the predicted survival rate of my patient, depending on the selection of treatment A or treatment B		two survival probabilities as derived from an AI algorithm (preferably including explanations)
paediatric cancer survivor	know which sources hold any data or samples of mine	have an overview of my data and samples	a table of data sources and types of data/samples they hold of mine
	receive life-long suggestions concerning screenings, etc., based on all my own data and recent study results	improve my life expectancy, health status, and quality of life	suggestions and summaries provided by a survivorship app
	make sure that my data are used for further research in the most valuable way	contribute to the improvement of paediatric cancer treatment	–
researcher in the field of PO	compare the outcome of treatment A with that of treatment B (taken from source A) in patients fulfilling a certain eligibility criterion (taken from source B)	see if the type of treatment is correlated with the outcome	a Kaplan–Meier-curve with two groups, for treatments A and B
	compare the outcome of study A with that of study B in patients fulfilling certain criteria	see if one of the studies’ outcomes is superior	a Kaplan–Meier curve with two groups, for studies A and B
	know all follow-up results of one of the patients in my study, no matter within which context the follow-up was performed	use the most recent follow-up data for the analysis of my study	a table of all follow-ups including dates, sources, and results
researcher in the field of biology	know which biobanks have further samples on which I need to perform an additional experiment	contact these biobanks and ask for additional material	a table with the sample type and biobank including contact information
	know specific results, as stored in genome–phenome analysis platforms, which were achieved with the probes from my biobank	gain further insights into the properties of my samples	a table with platforms and results per probe in my biobank
	know which sources contain patients with a specific biomarker	contact these sources to perform further research on that biomarker	a table with the number of patients per source
future principal investigator of a specific research activity	know how many cases fulfilling specific eligibility criteria are present in at least two data sources	estimate the number of cases for my study	a number of overall cases
	know which data sources contain cases fulfilling specific eligibility criteria (age, diagnosis, biomarker, treatment, etc.)	get in contact with these sources to consider them in my study	a table with cases per source
	correlate variables in patients from various sources, fulfilling my eligibility criteria, with one another	optimise stratification in my study	correlation between the variables
member of the SIOPE board	compare the survival rate of all neuroblastoma patients in all sources in 2001–2010 to the rate in 2011–2020	evaluate the improvements in neuroblastoma research in Europe	a boxplot
	compare the rate of severe adverse events per member state during immunotherapy with a certain biological in all sources	identify differences across Europe and improve potential shortfalls	a landscape, color-coded with the rate of severe adverse events
healthcare politician	know whether a specific type of PO treatment is cost-effective	optimise PO treatment based on outcomes and costs	table listing costs and outcomes of two different PO treatment options

## Data Availability

Data are contained within the article.

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
