# Peer review of "Use Cases Requiring Privacy-Preserving Record Linkage in Paediatric Oncology"

_cancers, 2024, doi:10.3390/cancers16152696_

Round 1
Reviewer 1 Report
Comments and Suggestions for Authors
Record linkage is crucial in clinical and epidemiological research. This article is (1) a good review of record linkage use-cases; (2) a good and didactic presentation of the landscape of records that are generated by patients along the health care system; and also (3) of the records linkage stages to profit the wealth of data generated in the clinical, laboratory, administrative and epidemiological settings.
The need of record linkage is increasing while the necessary data protection regulations put difficulties in its application. Therefore Privacy-Preserving Record Linkage (PPRL) methods, as presented in this article are of paramount importance.
References in general and self-citations are appropriate
Suggestions for the authors:
-A list of acronims and abbreviations would be useful. Please, introduce it.
-The number of the paragraphs in page 8, row 272 and page 9, row 322, seem wrong. Please, check them.
-In page 6, rows 169 and 172, change the term "Chapter" by the term "Section"
-Introduce some examples of linkage with population-based registries
-Finally, expand the end of the Discussion with a comment retaking the last paragraph in page 4 (rows 118-123) and your assessment of the tools available in Europe and the appropriate contacts.
Comments on the Quality of English Language
Some sentences carry implicit meaning. It would be useful for purposes of dissemination among a wider audience to be more explicit.
Author Response
Thank you very much for your kind review and for the valuable comments. Please find our answers below:
Comment 1: A list of acronims and abbreviations would be useful. Please, introduce it.
Response 1: Has been introduced.
Comment 2: The number of the paragraphs in page 8, row 272 and page 9, row 322, seem wrong. Please, check them.
Response 2: Has been corrected.
Comment 3: In page 6, rows 169 and 172, change the term "Chapter" by the term "Section"
Response 3: Has been changed.
Comment 4: Introduce some examples of linkage with population-based registries
Response 4: We have changed the section heading and added further PO examples in section 3.2.2 / Linked data stores <and registries>. Additionally, we have extended section 3.2.3 by providing more details on the cited papers and the use cases addressed.
Comment 5: Finally, expand the end of the Discussion with a comment retaking the last paragraph in page 4 (rows 118-123) and your assessment of the tools available in Europe and the appropriate contacts.
Response 5: The discussion has been expanded accordingly.
Finally, we edited the English language, corrected some typos and spelling errors and applied the MDPI reference format.
Reviewer 2 Report
Comments and Suggestions for Authors
The text entitled "Use Cases Requiring Privacy-Preserving Record Linkage in Pediatric Oncology" is very well written. The introductory part is concise and easily readable even for the reader not in the field. The explanatory section of the methods is well written as is the results part.
The authors should cite other collaborative studies, beyond the Neuroblastoma study, (including biological material, biobanks, short- and long-term patient outcomes).
The authors should also report how the collaborative study on Neuroblastoma has allowed (over the years) to eliminate randoms because the international data had highlighted a clear advantage of one arm over the other.
The English language is good.
Comments on the Quality of English LanguageThe English language is good.
Author Response
Thank you for your kind review and for the valuable comments. Please find our answers below.
Comment 1: The authors should cite other collaborative studies, beyond the Neuroblastoma study, (including biological material, biobanks, short- and long-term patient outcomes).
Response 1: We agree that there are various collaborative studies beyond Neuroblastoma that highlight the potential of PPRL in PO. We had mentioned some of the identified use cases in section 3.2.3. However, this section had obviously been too short. Therefore, we have extended this section by providing more details on the cited papers and the use cases addressed. Additionally, we have changed the section heading and added further PO examples in section 3.2.2 / Linked data stores <and registries>
Comment 2: The authors should also report how the collaborative study on Neuroblastoma has allowed (over the years) to eliminate randoms because the international data had highlighted a clear advantage of one arm over the other.
Response 2: A paragraph highlighting the improvements in neuroblastoma survival rates and the role of PPRL has been added to the discussion. 
Finally, we edited the English language, corrected some typos and spelling errors and applied the MDPI reference format.